# Biodiversity of marine microbes is safeguarded by phenotypic heterogeneity in ecological traits

**Susanne Menden-Deuer[1], Julie Rowlett[2]\*, Medet Nursultanov[3], Sinead Collins[4], Tatiana Rynearson[1]**

1 Graduate School of Oceanography, University of Rhode Island, Narragansett Bay Campus, Narragansett, RI, United States of America, 2 Mathematical Sciences, Chalmers University and the University of Gothenburg, Gothenburg, Sweden, 3 School of Mathematics and Statistics, University of Sydney, Camperdown, Australia, 4 Biological Sciences, Ashworth Laboratories, University of Edinburgh, Edinburgh, United Kingdom

* julie.rowlett@chalmers.se

## Abstract

Why, contrary to theoretical predictions, do marine microbe communities harbor tremendous phenotypic heterogeneity? How can so many marine microbe species competing in the same niche coexist? We discovered a unifying explanation for both phenomena by investigating a non-cooperative game that interpolates between individual-level competitions and species-level outcomes. We identified all equilibrium strategies of the game. These strategies represent the probability distribution of competitive abilities (e.g. traits) and are characterized by maximal phenotypic heterogeneity. They are also neutral towards each other in the sense that an unlimited number of species can co-exist while competing according to the equilibrium strategies. Whereas prior theory predicts that natural selection would minimize trait variation around an optimum value, here we obtained a mathematical proof that species with maximally variable traits are those that endure. This discrepancy may reflect a disparity between predictions from models developed for larger organisms in contrast to our microbe-centric model. Rigorous mathematics proves that phenotypic heterogeneity is itself a mechanistic underpinning of microbial diversity. This discovery has fundamental ramifications for microbial ecology and may represent an adaptive reservoir sheltering biodiversity in changing environmental conditions.

## Introduction

Marine planktonic microbes make Earth habitable by collectively generating as much organic matter and oxygen as all terrestrial plants combined [1]. The ecological, biogeochemical and economical importance of marine microbes is rooted in their vast species and metabolic diversity [1, 2]. Recent estimates of marine microbial species diversity exceed 200,000 species in the plankton [3, 4]. At all levels of taxonomy, from species to intra-strain comparisons, there exists a tremendous and theoretically inexplicable variability in genetic, physiological, behavioral

**Data Availability Statement:** This work is based on a mathematical proof that is contained in the Supporting information file.

**Funding:** SMD supported by National Aeronautics and Space Administration programs: EXport Processes in the global Ocean from RemoTe Sensing (EXPORTS) field campaign (grant 80NSSC17K0716). JR supported by the Swedish Research Council Grant 2018-03873. MN supported by the Australian Research Council Grants ARC DP190103302 and ARC DP190103451. Further support was provided by National Science Foundation awards 1736635 (SMD), 1638834 (TAR) and 1543245 (TAR). National Science Foundation award DMS-1440140 supported JR at the Mathematical Sciences Research Institute in Berkeley, California. None of the sponsors or funders played a role in any aspect of this manuscript or the research therein.

**Competing interests:** The authors have declared that no competing interests exist.

and morphological characteristics [5–17]. Recent methodological break throughs have allowed the discovery of heterogeneity at the level of gene expression activation and revealed rare subpopulations in bacteria [18]. The maintenance of such extraordinary diversity and persistent co-existence of planktonic microbes in a putatively isotropic environment represents a long-standing scientific enigma [19] that has yielded a phenomenologically sound explanation [20] but to date lacks a general, mechanistic explanation.

While community and intraspecific diversity is usually framed in terms of genotype diversity, genetically-identical cells often have important phenotypic differences in homogeneous environments [21–23], a phenomenon called phenotypic heterogeneity [24] or intra-genotypic variability [25]. Although phenotypic heterogeneity is a key attribute of microbes, it is not typically examined as a potential driver of microbial diversity or species persistence. Importantly, phenotypic heterogeneity can be acted on by natural selection; it is heritable, can be altered by genetic change, and can influence survival and fitness [21, 26, 27]. Previous treatments of phenotypic heterogeneity focus on how phenotypic heterogeneity can be generated [24, 28], how this characteristic can allow populations to use bet-hedging to persist in heterogenous environments [24], to support niche partitioning [20], or to facilitate intra-specific cooperation [29]. However, the role of phenotypic heterogeneity in maintaining species diversity remains unexplored.

We propose that phenotypic heterogeneity and incorporation of cell-cell interactions is key to understanding coexistence in microbial communities and explains how large numbers of species can coexist, even in unstructured environments. Here we examined how phenotypic heterogeneity affects population survival and coexistence. This corresponds to a scenario where all competing species are reasonably well-adapted to their environment. We leveraged game theory [30] as a tool to explore the consequences of phenotypic heterogeneity for the outcomes of cell-cell competitions at the individual level and the resulting implications for the persistence of populations. A non-cooperative game for competition between microbe species was introduced in [31] and [32]. In that model, a player in the game-theoretic-sense represents a species that is comprised of many individual microbes. This provides a crucial link between individual-level interactions and species-level repercussions. Importantly, the approach taken specifically formulated a model to reflect the characteristics of asexually reproducing microbes, and thus departs from approaches that utilize models designed for multicellular organisms to understand microbial ecology and evolution. In [32], competitions between pairs of species were analyzed, and it was shown that there are no evolutionary stable strategies. That result indicated that optimal trait distributions for microbe species might not be centered around an optimal value, but instead, that perhaps microbe species may be characterized by more variable trait distributions. However, the model in [32] did not analyze competition between arbitrary and fluctuating numbers of competing species, and in particular, did not identify the best strategies to promote survival and co-existence. Here, we generalize the model in [32] to create a non-cooperative game in which arbitrary and fluctuating numbers of microbe species compete. We identify all equilibrium strategies. These equilibrium strategies have two key characteristics: maximal phenotypic heterogeneity and neutrality towards all other strategies that are equally cumulatively fit. These findings represent a remarkable departure from current conceptualization and modeling centered around fitness optimization and competitive exclusion.

## Methods: A microbe centric competition model that links individual-level interactions to population-level consequences

Taking a microbe-centric point of view, we assume that species are represented by many individual, closely-related cells. These individuals compete for limited resources, to avoid

## Two cells compete

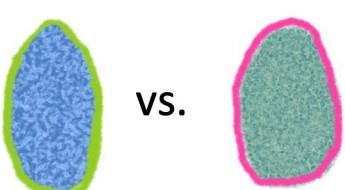

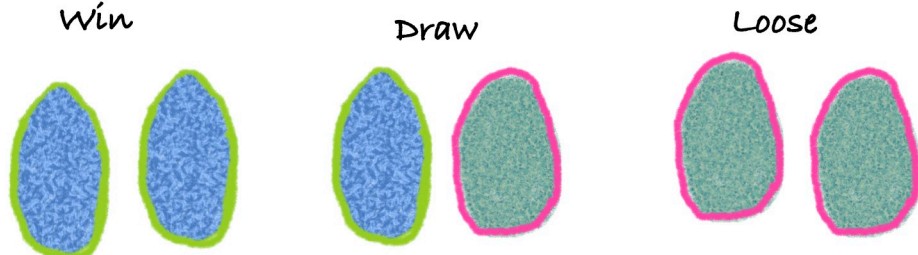

| P2 \ P1 | W | L |
|---|---|---|
| W | (0,0) | (1,-1) |
| L | (-1,1) | (0,0) |

## 3 possible outcomes

**Fig 1. Competition occurs on a cell-to-cell level.** When two asexually reproducing individuals compete in a zero-sum game, there are three possible outcomes for each individual: win = self-replicate; draw = maintain status quo for each competitor; lose = die/competitor replicates. The table shows the pay-off function of individual-level competition. A payoff of 1 corresponds to win, a payoff of -1 corresponds to lose, and a payoff of 0 corresponds to a draw.

predation, and other, ecologically relevant factors. The cumulative outcome of competition between individuals, no matter how biologically complex, can be reduced to three possibilities: win, lose, or draw. Since microorganisms reproduce asexually, success or failure in competition corresponds directly to population increase or decrease, respectively, as shown in Fig 1.

At each round of competition, each individual is assigned a competitive ability according to the strategy of its species. Biologically, a strategy represents the probability distribution of competitive abilities (e.g. traits) for all individuals of the species. Different trait distributions can be the result of random phenotypic noise, plasticity, or the genetic variation that inevitably exists in large, actively-dividing microbial populations [33]. Leveraging theoretical mathematics, we are able to simultaneously consider all traits and all types of distributions, rather than single traits with close to normal distributions as done previously [34]. In the game theoretic sense, a trait distribution is a mixed strategy for the species, considered as a player in a non-cooperative game. Representing competitive ability as a distribution, rather than a mean value exemplifies our approach of incorporating phenotypic heterogeneity in the assessment of competition outcomes.

Game theory is uniquely suited to examine the effects of phenotypic heterogeneity on survival of competing microbial species because game theory evaluates population-level outcomes of individual-level interactions as depicted in Fig 2. In our application of game theory, we define species-level strategies from which we derive the outcomes of individual-level competitions that ultimately determine the population-level survival and abundance. This approach is in contrast with population-level assessments, like Lotka-Volterra type models, where phenotypic heterogeneity is accounted for by using stochastic differential equations [35]. The game theoretic framework provides a quantitative means to evaluate the success of species competing according to their strategies. Below, we obtain tractable equations for each individual

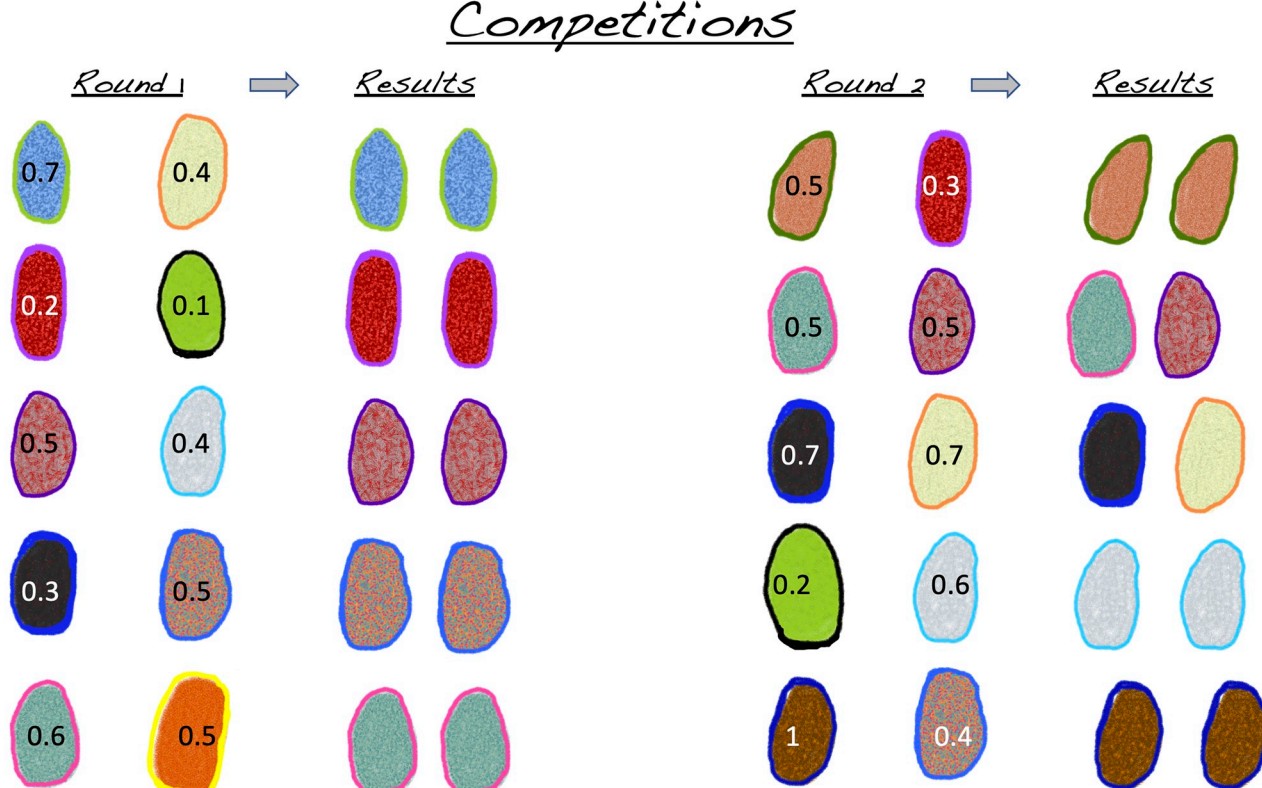

**Fig 2. Phenotypically heterogeneous individuals compete.** At each round of competition, a subset of diverse individuals from a large cohort competes and the outcomes of competition are assessed based on their relative competitive abilities (values within cells). Supported by empirical observations, the competitive ability of clonal individuals can be expressed heterogeneously in identical conditions and vary over time. Our work reflects this by assigning individuals their competitive ability according to the strategy of the species. Thus, variability amongst individuals and the phenotypically heterogeneous strategy of the species is preserved.

microbe, allowing us to mathematically represent individuals yet analyze all individuals in a population collectively. The resulting set of equations explicitly interpolates between microscopic or individual-interactions and macroscopic, or population-level consequences. A key realization is that there are many unique strategies that all have an identical mean competitive ability. This characteristic builds a fundament of why many species with phenotypic heterogeneity can co-exist: because the competing individuals can yet differ.

### The discrete and continuous models

We consider $n$ competing species and allow $n$ to vary. Each species has a strategy, and we may use the same notation to denote both the species as well as its strategy. Here we shall analyze two models: discrete and continuous. In the discrete model, there is a finite set of competitive abilities:

$$\left\{ \frac{j}{M} \right\}_{j=0}^{M}, \quad x_j := \frac{j}{M}.$$

**Definition 1**. *A strategy in the discrete case is a map $B : \{x_j\}_{j=0}^{M} \to [0, \infty)^{M+1}$ such that*

$$|B| := \sum_{j=0}^{M} B(x_j) > 0 \ and \ \mathrm{MCA}\,(B) := \frac{1}{|B|} \sum_{j=0}^{M} x_j B(x_j) \leq \frac{1}{2}. \tag{1}$$

*We use B to denote both the species and its strategy. The limitation on the* mean competitive ability (MCA) *reflects biological constraints, in the sense that individuals are not perfect.*

Consequently, biologically we interpret $|B|$ as the population abundance of $B$. A species whose population abundance is zero cannot compete; thus we only analyze strategies with a positive population abundance, but we note that the population abundance need not be integer-valued. The strategy assigns competitive abilities to individuals in the sense that $\frac{B(x_j)}{|B|}$ is the probability that a randomly selected individual from species $B$ has competitive ability equal to $x_j$. In game theory, it may seem more natural to define the strategy instead as a map

$$\tilde{B} : \{x_j\}_{j=0}^{M} \to [0, 1]^{M+1} \ \text{and require that} \sum_{j=0}^{M} \tilde{B}(x_j) = 1.$$

This would then imply that all species have population abundances equal to one, but we wish to allow species of different population sizes to compete, and therefore we do not make this normalization. However, it is wholly equivalent to consider $\tilde{B} := B/|B|$ as the strategy of the species with the population equal to $|B|$, requiring that $|B| > 0$ but need not equal one, and with this normalization indeed

$$\tilde{B} := \frac{B}{|B|} : \{x_j\}_{j=0}^{M} \to [0, 1]^{M+1}, \quad \sum_{j=0}^{M} \tilde{B}(x_j) = 1.$$

One could imagine that as species evolve, there are more and more different competitive abilities in the range from 0 to 1, and for this reason our analyses include a second model, *the continuous model*. In the continuous model, the competitive abilities are selected from the entire range of real numbers between 0 and 1. A species is associated with a continuous non-negative function defined on the unit interval, that is used to define the strategy of the species.

**Definition 2**. *A strategy in the continuous model is a continuous map $f: [0, 1] \to [0, \infty)$ that satisfies*

$$F(1) := \int_0^1 f(x)dx > 0 \ and \ \mathrm{MCA}\,(f) := \frac{1}{F(1)} \int_0^1 xf(x)dx \leq \frac{1}{2}. \tag{2}$$

Similar to the discrete model, a species whose population abundance is zero cannot compete, so we only analyze those species that have positive, but not necessarily integer-valued, population sizes. The continuous model is obtained from the discrete model by letting the number of competitive abilities $M \to \infty$. In game theory, it may seem more natural to define the strategy instead as the function

$$\tilde{f}(x) := \frac{f(x)}{F(1)} \Rightarrow \tilde{f}(x) \geq 0 \forall x, \quad \int_0^1 \tilde{f}(x)dx = 1.$$

The function $\tilde{f}$ is then the probability distribution of competitive abilities of the species, but with this normalization, again all species would have population abundances equal to one, and we wish to allow species of different population sizes to compete. Consequently, we do not

make this normalization, but it is mathematically equivalent to consider $\tilde{f}$ as the strategy of the species with the population to $F(1)$, requiring only that $F(1) > 0$ but need not equal one.

The change in population abundance of the species is determined by the game theoretic payoffs. These payoffs are defined by the expected value in competition when all species simultaneously compete. Consequently, in the discrete case, we define the payoff to species $A_k$ to be

$$\wp(A_k; A_1, \ldots, A_{k-1}, A_{k+1}, A_n) :=$$
$$\frac{1}{\sum_{\ell=1}^{n} |A_\ell|} \sum_{j=0}^{M} A_k(x_j) \left[ \sum_{i<j} \sum_{\ell=1}^{n} A_\ell(x_i) - \sum_{i>j} \sum_{\ell=1}^{n} A_\ell(x_i) \right]. \tag{3}$$

Above and throughout this work, an empty sum is taken to be equal to zero. Here, species are competing both internally and externally to the population. Due to the zero sum dynamic, however, internal competition within a species does not affect its population abundance. Consequently, these payoffs show how the species' population sizes increase, decrease, or remain stable when the species compete according to their strategies. The payoffs in the continuous case are defined in an analogous way, so that the payoff to species $f_k$ is

$$\wp(f_k; f_1, \ldots, f_{k-1}, f_{k+1}, \ldots, f_n) :=$$
$$\frac{1}{\sum_{\ell=1}^{n} F_\ell(1)} \int_0^1 f_k(x) \left[ \int_0^x \sum_{\ell=1}^{n} f_\ell(t) - \int_x^1 \sum_{\ell=1}^{n} f_\ell(t) \right] dx. \tag{4}$$

Having obtained this unique set of equations for all individual microbes connecting individual-level interactions to population-level consequences, we are poised to mathematically analyze all strategies. What are the best strategies? The answer to this question is a fundamental concept in non-cooperative game theory known as *an equilibrium point*. An equilibrium point is a collection of strategies for all players of the game such that no one player can increase their payoff if only that one player changes their strategy whilst other players keep their strategies fixed [36]. The strategies that comprise an equilibrium point are *equilibrium strategies*.

**Definition 3**. *In the discrete case, an* equilibrium point *is a set of strategies* $A_1, \ldots, A_n$ *as in Definition 1 such that for all* $k = 1, \ldots, n$ *we have*

$$\wp(A_k; A_1, \ldots, A_{k-1}, A_{k+1}, \ldots A_n) \geq \wp(B; A_1, \ldots, A_{k-1}, A_{k+1}, \ldots, A_n),$$

*for all strategies B as in Definition 1. In the continuous case, an* equilibrium point *is a set of strategies* $f_1, \ldots f_n$ *as in Definition 2 such that for all* $k = 1, \ldots, n$ *we have*

$$\wp(f_k; f_1, \ldots, f_{k-1}, f_{k+1}, \ldots f_n) \geq \wp(g; f_1, \ldots, f_{k-1}, f_{k+1}, \ldots, f_n),$$

*for all strategies g as in Definition 2. In both cases, the strategies that comprise an equilibrium point are* equilibrium strategies.

## Simulations for the discrete model

To visualize the theoretical mathematics, we simulated competitions for the discrete model, following the model described in [31]. Competition outcomes are based on a value for the 'competitive ability (CA)' which is a number between 0 and 1, with zero indicating poor competitive ability and 1 indicating an unbeatable winner. The competitive ability reflects fitness with respect to an ecologically meaningful trait of the individual, such as physiology, predator defence, or morphology. We view this CA as a cumulative trait, because a single competitive interaction does not typically yield as decisive an outcome as division or death. Thus our competitions reflect the cumulative outcomes of competitions over an organism's generation time.

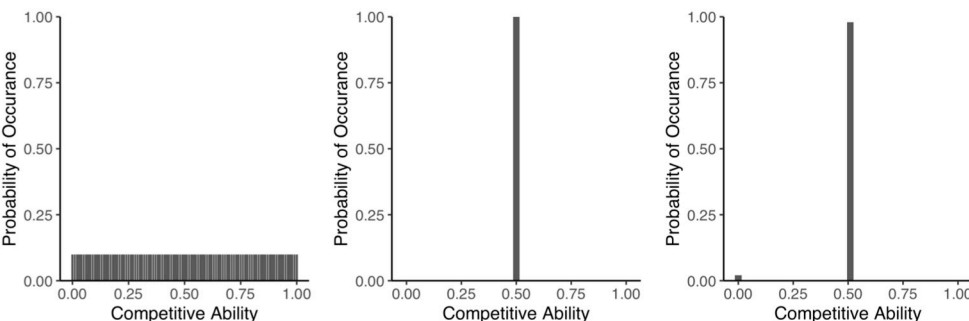

**Fig 3. Examples of strategies.** The strategies are represented by probability distributions that correspond to the distribution of competitive abilities (e.g. traits). A uniform distribution has all values between zero and 1 equally likely. An invariant or degenerate distribution is the opposite; all competitive abilities are equal to the mean lacking any variability. Lake Wobegon is characterized by one zero (loser) and all others have a mean slightly above 0.5, depending on population size. Our results show that the maximally variability, uniform distribution, is an equilibrium strategy that promotes co-existence.

The key is that the CA can be distributed in infinite ways and yet yield the same mean competitive ability. Some of the distributions we highlight here are illustrated in Fig 3.

The seemingly infinite variation in traits empirically observed (see citations in the Introduction) is implemented in this model by assigning cohorts of individuals (populations or species) infinite variability as characterized by the uniform distribution. The ramification of this population-level competitive strategy is contrasted with other commonly used strategies, such as a biomodal distribution or an invariant distribution (lacking variance represented by a single, constant competitive ability). Heterogeneous expression of traits even by clonal individuals [25] in identical environmental conditions is manifested in the model by allowing random assignments of competitive abilities (CA) to individuals, and allowing those to change over time, irrespective of prior successes of that individual. Ecologically, this reflects conditions where one trait may be advantageous in one condition but disadvantageous in another condition. Each competition is evaluated discretely between two individuals.

## Results

Our main result is Theorem 1, that presents all equilibrium strategies for any number of competing species. The theorem is visualized by simulations that are consistent with the theorem. These simulations visualize the theorem for a subset of cases, although the mathematical theorem holds for all infinitely many cases.

**Theorem 1**. *In the discrete case, assume that the set of competitive abilities is $\{x_j = \frac{j}{M}\}_{j=0}^{M}$. If M is odd, then equilibrium strategies are precisely all those that have positive and identical values at $x_j$ for all j. If M is even, then B is an equilibrium strategy if and only if $\mathrm{MCA}(B) = \frac{1}{2}$, and B $(x_{2j}) = B(x_0)$, and $B(x_{2j+1}) = B(x_1)$ for $j = 0, \ldots, \frac{M}{2}$. In the continuous case, a strategy is an equilibrium strategy if and only if it is a positive constant function. In both models, in all cases, the sum of two or more equilibrium strategies is an equilibrium strategy. In both models, in all cases, any collection of equilibrium strategies is an equilibrium point, and conversely, every equilibrium point is comprised of these equilibrium strategies.*

An equilibrium strategy indicates what type of competitive ability distributions one may expect to find in nature. Interestingly, the equilibrium strategies assuring species persistence are all characterized by maximal phenotypic heterogeneity, that is species whose competitive abilities cover the whole range of possible trait values, rather than a restricted range of values

centered around a particular value. Theorem 1 also shows that combining strategies of species that are characterized by equilibrium strategies, the result is still an equilibrium strategy.

Equilibrium strategies are characterized by maximal phenotypic heterogeneity in the sense that the competitive abilities are spread over the entire range of possible values rather than clustered around the mean. For any strategy that is not an equilibrium strategy, in (S1 and S2 Appendices) for the discrete and continuous models, respectively, we design a competitor strategy that is cumulatively equally fit, yet due to the particular features of its trait distribution can outcompete the non-equilibrium strategy, leading to the extinction of the less variable competitor. For example, in the discrete model an invariant or degenerate strategy, lacking variability, has a probability of 0 for every competitive ability not equal to the mean. One of many strategies that will defeat the invariant strategy is Lake Wobegon. (We named this 'designer distribution' Lake Wobegon based on a North American radio show, the Prairie Home Companion. Although the mean competitive ability is 0.5 like the other strategies, for this strategy, one individual has a competitive ability of 0 which results in all others having a slightly above average competitive ability, depending on population size. As the saying in the show went, 'all the children (except one) are above average.') Such a strategy assigns the majority of individuals' competitive ability slightly above the mean competitive ability, while one individual is assigned competitive ability 0, in this way guaranteeing that the mean competitive ability of the Lake Wobegon strategy is equal to that of the invariant strategy. Similarly, we give a recipe for constructing a superior strategy to outcompete *any* non-equilibrium strategy in the aforementioned sections of the supporting information. We do not claim that these designer strategies reflect biological realism. Instead, they demonstrate that the equilibrium strategies are superior in their ability to coexist with any other strategy and resist replacement through competition or invasion.

Having identified the key role of phenotypic heterogeneity in species competition and coexistence, we explored the ramifications of specific distributions of competitive strategies commonly used in ecological models, such as gaussian, bimodal, and degenerate distributions. These types of distributions are often assumed to underlie the metric under study [37]. To visualize the effects of specific strategies representing different degrees of phenotypic heterogeneity on the outcomes of competition, persistence and invasion, we utilized an individual-based competition simulation that emulates the rules of our game theoretic approach [31]. Groups of individuals, which can represent populations or species, compete in a randomized design as depicted in Fig 2. These competitions are based on the discrete model. We assume no heritability of a specific competitive ability across generations and rather invoke the maintenance of the strategy at the species level. This assumption is well supported by empirical observations that individuals modulate key characteristics, including variations in growth rate, as a function of the presence of conspecifics or competing species [16, 38]. This shows that rather than traits that are strictly executed based on environmental conditions, traits are further modulated by which species or individuals are present. The outcomes of these competitions on population abundance are evaluated in an individual based model as shown in Figs 4–6. In each instance, the simulations reproduce the exact outcomes predicted by the mathematically-derived payoff functions. These simulations provide a visualization of the game theoretic model for the discrete case.

Most notably, our theory predicts that equilibrium strategies are capable of co-existing with any other strategy that has the same mean competitive ability and cannot be replaced by another strategy that has the equal or lower mean competitive ability. The predicted co-existence of the maximally variable, uniform distribution is illustrated for some competitor strategies in Fig 7. The theory also predicts that a population characterized by an equilibrium strategy starting orders of magnitude lower in abundance than its competitors can persist,

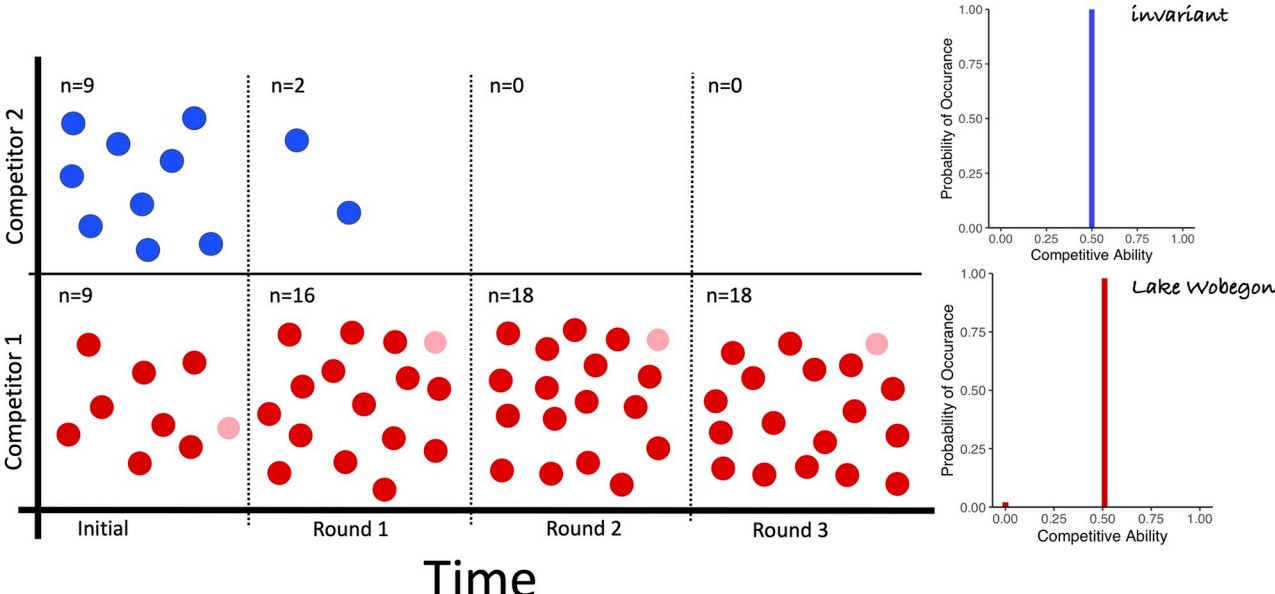

**Fig 4. Phenotypic heterogeneity persists whereas invariability can be eliminated.** Three rounds of competition are simulated for two strategies. Probability distributions of strategies are illustrated on the right with individuals (n) represented by color coded circles. Here the invariant strategy (top, blue), characterized by lack of variability and a constant competitive ability equal to the mean, is eliminated by the Lake Wobegon strategy (bottom, red), characterized by one individual with 0 competitive ability and all others are 'above average' (mean+x for a small $x > 0$). This and the following figures show few individuals for clarity but simulations utilized larger sample sizes for accurate representation of the probability distributions of competitive abilities.

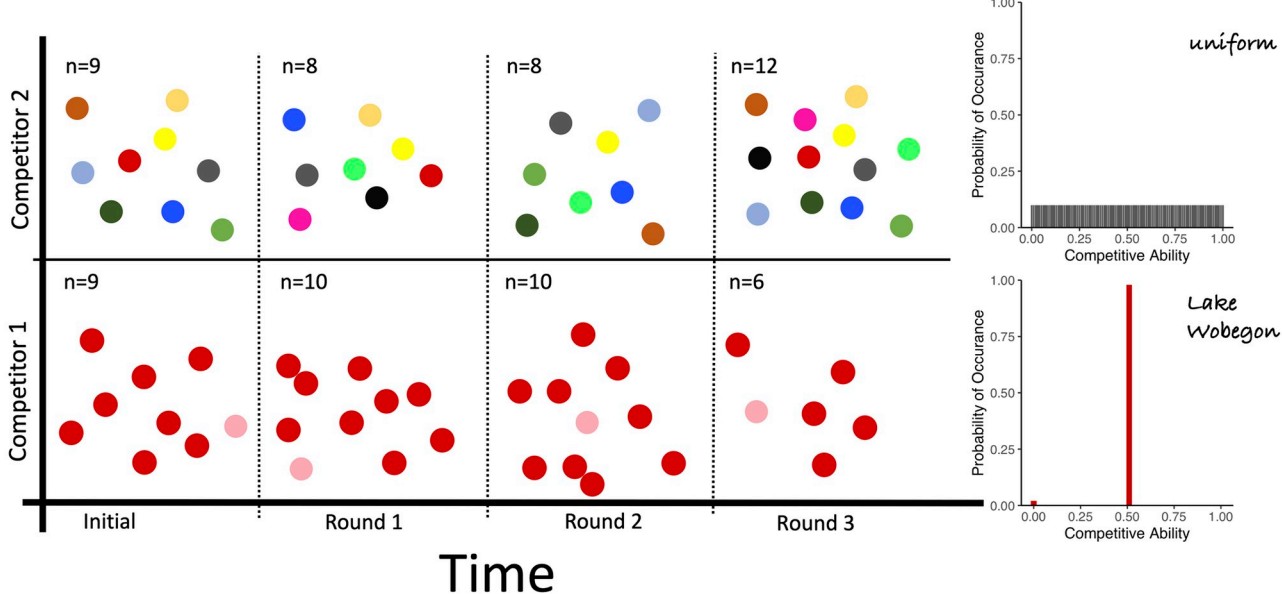

**Fig 5. Phenotypic heterogeneity co-exists with the strategy that defeats invariant.** In the same type of three time-step competition as in Fig 4, the Lake Wobegon strategy that defeats invariant coexists with an equilibrium strategy (top, multicolored). Note the rainbow color range for the equilibrium strategy reflects its maximal phenotypic variability.

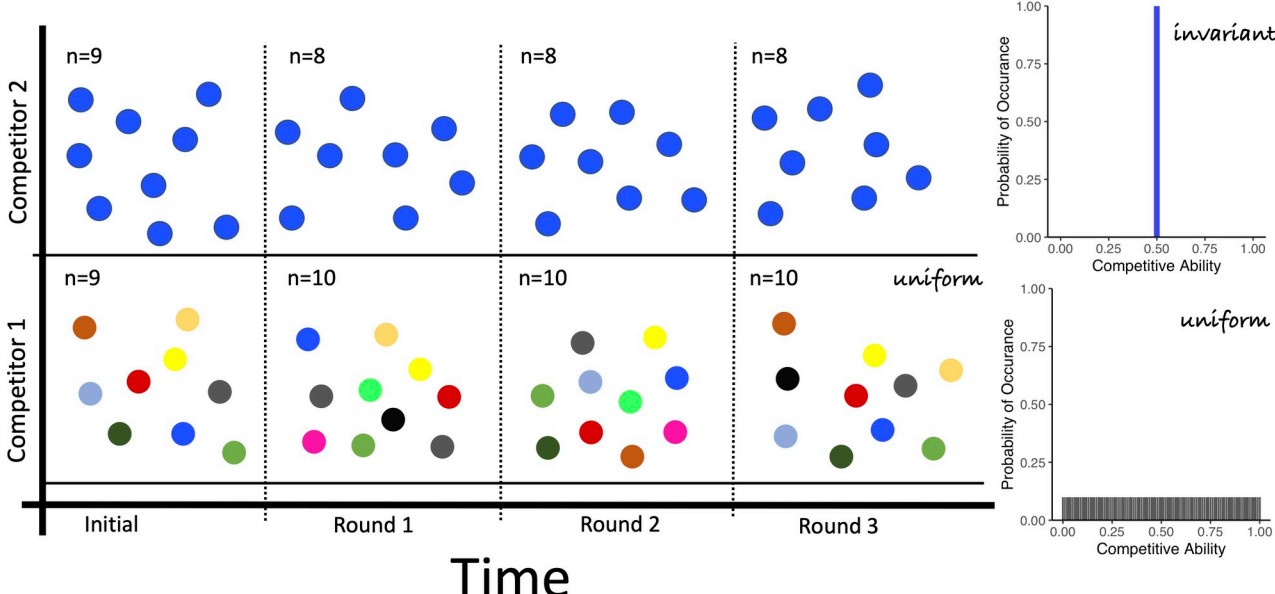

**Fig 6. Phenotypically heterogeneous strategies enable co-existence.** An equilibrium strategy also co-exists with the invariant strategy and in general co-exists with all strategies that have the same mean competitive ability.

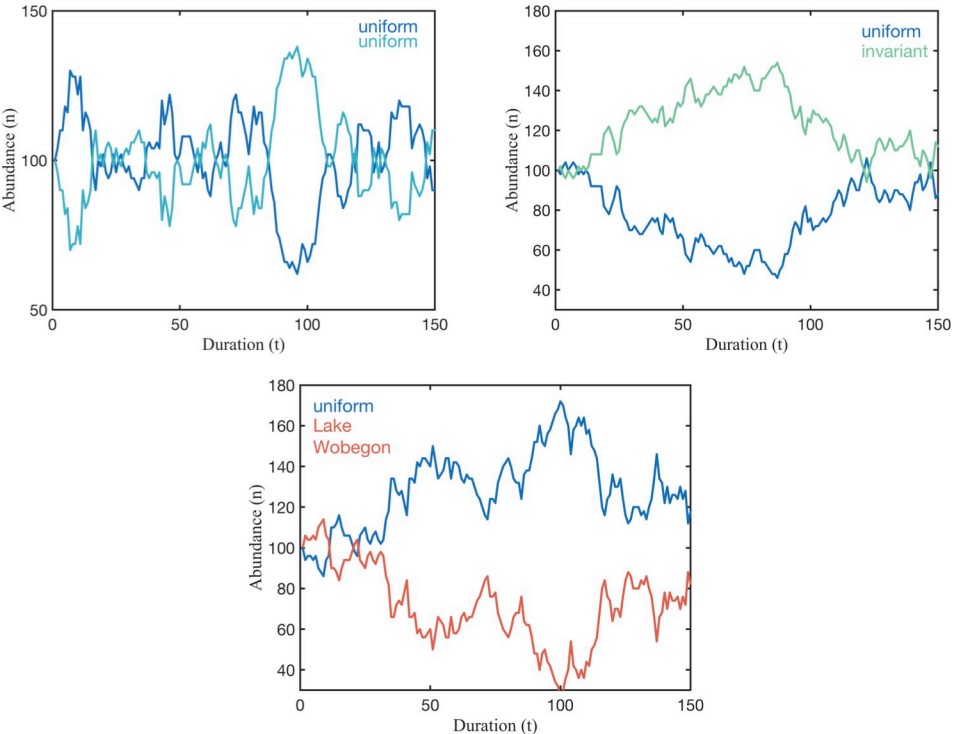

**Fig 7. Equilibrium strategies resist invasion.** Simulation results for larger population abundances and longer durations confirm the examples shown in the visualizations in Figs 4 through 6. A uniform distribution is an equilibrium strategy. It co-exists with all strategies that have the same mean competitive ability.

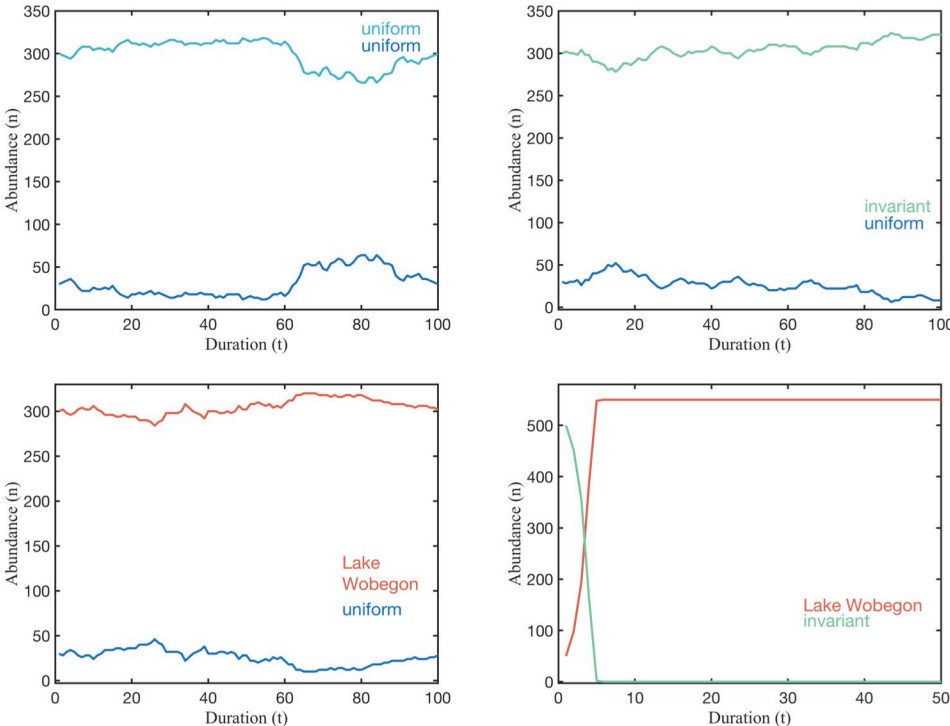

**Fig 8. Phenotypic heterogeneity promotes infinite co-existence and thwarts invasion.** The uniform distribution, an equilibrium strategy, persists even when their population abudance is vastly lower than their competitor's. In contrast, the invariant strategy starting at a much higher population is quickly replaced by the Lake Wobegon strategy. Note that the mean competitive abilities of all strategies are identical.

while an invariant population at high abundances can be replaced. Again the simulations display the theoretical predictions as shown in Fig 8. The robustness of the equilibrium strategies starkly contrasts the weakness of the invariant strategy, where a population is characterized only by its average competitive ability with no variability about the mean. Remarkably, several strategies can be identified that can replace the invariant strategy as shown in Fig 8, suggesting that the invariant strategy should not persist when other populations of equal mean fitness are present. Thus, a common practice of representing empirical measurements of, for example, microbial physiology as averages may mask the important information of variance in physiological capacity. Population size, with exception of very small population size as discussed in (S1 Fig), and duration does not affect these results. Naturally, there are cases where extinctions are observed as shown in (S1 Fig). Our simulations faithfully reproduce the predictions of the competitive exclusion principle [39] when competition involves an overall inferior competitor, characterized by a lower mean competitive ability; see (S1 Fig). Extinctions of equilibrium strategies are observed in cases when the population size is very low (10s of individuals) because equilibrium strategy distributions cannot faithfully be reproduced when few individuals represent it; see (S2 Fig). This is consistent with chance, rather than selection, dominating persistence for very small populations. Aside from these explainable deviations, the finding that a maximally-variable distribution of ecological traits will persist in competition has remarkable implications for the ecology and evolution of free living microbes and how we should study them. Maximal intra-specific variability supports unlimited co-existence of species with equal mean fitness, as demonstrated in a simulation with 100s of populations, all represented by maximally intra-specific strategies as shown in Fig 9.

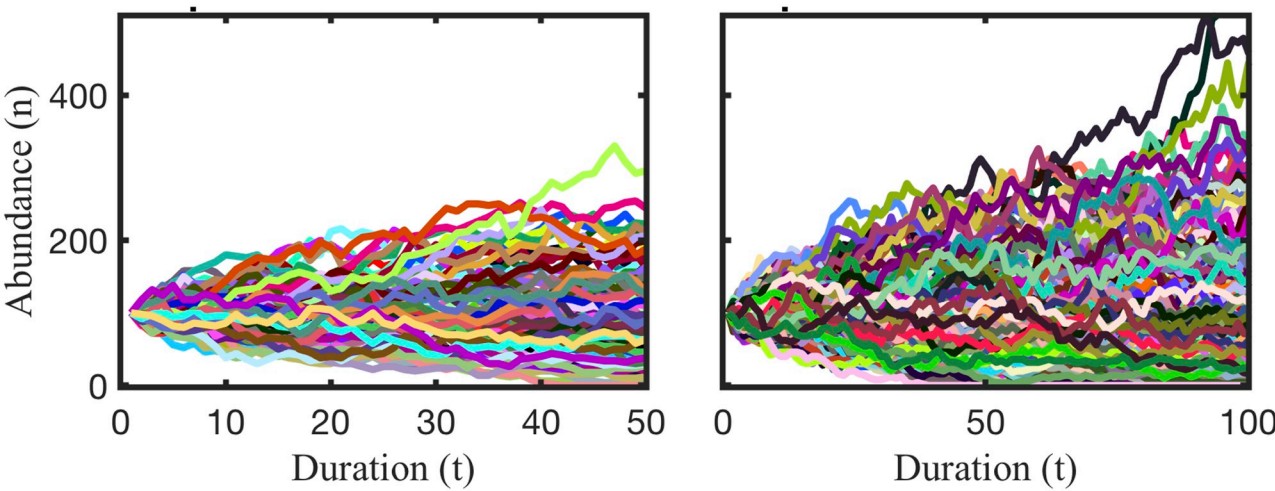

**Fig 9. For phenotypically heterogeneous populations, an unlimited number can co-exist.** Abundance of 101 (left) and 301 (right) equilibrium strategies in 50 to 100 rounds of competition. Although abundance fluctuates over time and no one population dominates, there are less than a handful of observable extinctions. The number of co-existing species displayed is limited by CPUs and visibility. Theoretically, the number is infinite.

## Discussion

Here we have obtained a rigorous mathematical proof that the best strategies for competing species have the maximum possible mean competitive ability, maximal phenotypic heterogeneity, and will defeat all strategies that have lower mean competitive ability, as one would expect. Surprisingly, these strategies are *neutral* towards all other species that also have a strategy with the maximal mean competitive ability. An immediate consequence is that there are no evolutionary stable strategies; this is consistent with the results of [32].

### A comparison of equilibrium strategies and evolutionary stable strategies

The distinction between evolutionary stable strategies (ESS) and equilibrium strategies is that whereas an ESS is the best strategy to face a specific challenge, equilibrium strategies should be understood as the best strategies to simultaneously face all challenges. For example, a specific ecological challenge faced by many photosynthetic microbes is balancing the tradeoff for light and nutrients that are available in opposite concentration gradients in surface waters of lakes and oceans. Although the optimal depth that best balances the tradeoff for light and nutrient resources may be an evolutionary stable strategy [40], that does not apply when conditions change, such as when predators appear. Algal aggregations within a single depth or a patch have been shown to be necessary for survival of zooplankton predators [41], and empirical research has supported this prediction [42]. In general, plankton inhabit a patchy environment and contribute to environmental heterogeneity [43]. So, while an ESS for a particular scenario of constraints may be found, such as for resource acquisition, it would not apply to other ecological challenges like predator avoidance.

Considering the heterogeneous and dynamic environment inhabited especially by marine microbes, the lack of an evolutionary stable strategy [32] is logical and drives the need for a more general concept. Accommodating the specific characteristics of microbial populations, we analyzed the fitness of species by interpolating between individual-level competitions and species-level population growth or decline. Previous work has conceptualized species coexistence in microbial communities by invoking spatial or temporal heterogeneity in some form [44–46]. Consequently, when placed in a common environment, species must have different

mean fitness, in the sense that they must be locally adapted to different environmental niches. In contrast, our theory shows that multiple species with the same average fitness in the same environment (niche) can coexist indefinitely, as long as individuals within those species vary maximally in their competitive abilities. The theory predicts survival of the most variable and equally cumulatively fit individuals and thus predicts co-existence of such species. This finding implies that selection in generally adapted species favors maintenance of variability rather than the evolution of optimized trait values but also that the production of intra-specific or intra-genotype variation in trait values is adaptive. The genetic maintenance of intra-genomic variability has been reported [25]. The distribution of risk afforded by phenotypic heterogeneity also points to a potential adaptive reservoir in physiological, behavioral and genetic diversity in light of large-scale changes occurring in ecosystems, including climate change [47, 48].

## Intra-specific phenotypic variability thwarts invasion, promotes co-existence, and allows emergence of new species

These mathematical proofs and supporting model results provide a fundamentally different mechanism for species co-existence than those previously advanced. There are several prior demonstrations that multiple species can coexist in the same niche or consuming the same resource, thus circumventing the competitive exclusion principle, such as two predators existing on different life stages of the same prey species, or asynchronous introduction of two competitors into a system; see [44, 49] and references therein. These previously advanced solutions are similar in that they introduce some asynchrony in the competing species, thus essentially separating competitors and eliminating direct competition. This is equivalent to invoking environmental patchiness in either space or time to increase the number of available niches. This type of asynchrony likely enhances species co-existence in numerous cases but is fundamentally an extension of the competitive exclusion principle; it expands the number of niches but still relies on an assumption of one species per niche. Our results agree with the two species model in [34] but contradict their suggestion that in multi-species competition, phenotypic heterogeneity depresses biodiversity. There is no mathematical contradiction because [34] considered a single trait with variations around a normal distribution whereas we mathematically analyze all traits and all distributions. The mechanism advanced here directly confronts competitors and is flexible in terms of the assumptions regarding population sizes and other specifics. We show that 100's of species can coexist in the same niche despite direct cell-cell competitions simply by incorporating the high phenotypic heterogeneity that is one of the empirically observed, fundamental characteristics of microbial populations.

This study also identifies a novel adaptive advantage (persistence in the face of competitors) and consequence (maintenance of diversity) of phenotypic heterogeneity. Phenotypic heterogeneity has previously been suggested to be adaptive because it can increase survival in heterogenous or fluctuating environments by allowing genotypes to bet-hedge [24], but there are no studies addressing how phenotypic heterogeneity may be advantageous in a homogeneous environment, and why natural selection might favor a particular shape of phenotypic heterogeneity. We demonstrate that because competitive interactions occur between individuals, the heterogenous condition can just as easily stem from the phenotypic heterogeneity in capabilities of competitors rather than features of the abiotic environment. Our findings explain how very high levels of genotype diversity [11, 50] can be maintained without invoking niche partitioning within species.

## Conclusion: A paradigm shift is needed to adequately conceptualize microbial populations

Our theory implies that a shift in perspective is needed when conceptualizing microbial populations. A microbe-centric perspective is necessitated by the fact that microbial ecology and evolution are subject to very different principles than the multi-cellular organisms upon which much biological theory was developed. The fundamental importance of microbes requires that the shape of the distribution of competitive strategies be investigated as a possible unifying and structuring principle governing the biodiversity of asexually reproducing microbes. We predict that maximally variable trait distributions will be typical, because of their adaptive features. Our results also suggest that high phenotypic heterogeneity is expected in populations that persist through multiple rounds of competition even in the absence of environmental patchiness and that such heterogeneity may largely reflect inherent characteristics of species [51] rather than methodological limitations or differences [8]. Interestingly, the discovered equilibrium strategies display the very same features observed in microbes: (1) these strategies have maximal phenotypic heterogeneity and (2) there are an unlimited number of these strategies which are neutral towards each other, implying they cannot universally competitively exclude each other. The mathematics reflects the biological qualities of microbes. This fundamental model for microbes can accommodate more complicated variations, which we have not explored here in order to convey the fundaments of the theory.

Disease dynamics, ecosystem function and evolutionary insights can be propelled forward by accounting for the direct linkage from individuals to communities to macroscopic processes such as the abundance and distribution of organisms without the loss of insight imparted by averaging out variability. We suggest that inherent phenotypic heterogeneity may be an essential determinant of microbe ecology. Remarkably, the role of phenotypic heterogeneity is in direct contrast to diversity maintenance mechanisms that rely on ever-increasing numbers of niches because of spatial and temporal heterogeneity [46, 52], or more inclusive lists of traits that underlie fitness [53]. If true, our theory suggests a fundamentally different mechanism, one where trait selection favors variable distributions, rather than an optimum value with minimal variation. Furthermore, the distribution is at least as important as the average. There should be stabilizing selection on the generation of phenotypic heterogeneity in an adapted population, rather than a minimization of that variation around an optimum value.

A major implication of these findings is the prediction that entropy in the system always increases because equilibrium strategies have maximal entropy and promote unlimited coexistence and emergence of new species. Biological diversity continuously expands. It has not escaped our notice that we may have also discovered a mechanism to explain one of the fundamental laws in biology, that of ever-increasing complexity in evolving systems [54]. Our focus here was chiefly on marine microbes because we can support our arguments with empirical observations and constrain descriptions to specifics. Yet, there is no reason why the same rules should not apply to other, free-living microbes. Rather than habitat, phenotypic heterogeneity is a key factor in species persistence and diversity. Based on the mathematics, there is nothing to suggest that other free-living microbes would not be subject to similar principles. In fact, we believe that here we present a unifying theory that phenotypic heterogeneity enhances biodiversity.

## Supporting information

**S1 Fig. Reproducing competitive exclusion.** Competition between uniform (blue) and an inferior competitor (orange) that has mean competitive ability < 0.5. In this case, our model

faithfully reproduces the 'competitive exclusion' principle [39], and the species with the lower mean is eliminated.
(TIF)

**S2 Fig. Low population abundances may not reproduce phenotypic heterogeneity.** On the left, the nominally uniform distribution goes extinct when the uniform distribution is statistically poorly characterized because few individuals represent the distribution. The right shows the mean competitive ability of the two cohorts in this simulation. The inferior mean in time step 3 results in extinction of the blue type. Even a reversal of fortunes in the next time step, with a higher mean CA for blue than orange does not balance the competition, and the blue population is eliminated. The blue-uniform strategy could have persisted longer by chance alone, but the randomly selected orange competitors all had superior competitive ability due to low population abundances. The trivial example of competition with an inferior competitor, characterized by a lower mean competitive ability, leads to extinction as shown in S1 Fig. At low population sizes the underlying strategy distributions can be poorly represented, especially for the maximally variable uniform distribution as shown in S2 Fig. In these cases, extinctions can be observed. Such dynamics would apply when few individuals colonize a new habitat, mutations yield new functions or other events when a few individuals represent the entirety of the population.
(TIF)

**S1 Appendix. Mathematical proofs for the discrete model.**
(ZIP)

**S2 Appendix. Mathematical proofs for the continuous model.**
(ZIP)

## Acknowledgments

We thank C.-J. Karlsson and B. Chen for providing constructive criticism on a preliminary draft of this manuscript and H. McNair and P. Marrec for help with the figures. Insights provided through the review process helped to improve the quality of the manuscript, for which we are very grateful.

## Author Contributions

**Conceptualization:** Susanne Menden-Deuer, Julie Rowlett.

**Formal analysis:** Susanne Menden-Deuer, Julie Rowlett.

**Funding acquisition:** Susanne Menden-Deuer, Julie Rowlett.

**Investigation:** Susanne Menden-Deuer, Julie Rowlett.

**Methodology:** Susanne Menden-Deuer, Julie Rowlett.

**Validation:** Susanne Menden-Deuer.

**Visualization:** Susanne Menden-Deuer, Julie Rowlett.

**Writing – original draft:** Susanne Menden-Deuer, Julie Rowlett, Medet Nursultanov.

**Writing – review & editing:** Susanne Menden-Deuer, Julie Rowlett, Medet Nursultanov, Sinead Collins, Tatiana Rynearson.

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
