## [Decision Letter · Decision Letter 0]

23 Apr 2021

PONE-D-21-01716

Biodiversity of marine microbes is safeguarded by phenotypic variability in ecological traits

PLOS ONE

Dear Dr. Rowlett,

Thank you for submitting your manuscript to PLOS ONE. After careful consideration, we feel that it has merit but does not fully meet PLOS ONE’s publication criteria as it currently stands. Therefore, we invite you to submit a revised version of the manuscript that addresses the points raised during the review process.

The manuscript provides an interesting insight in the field of microbial ecology and I tend to share the views expressed by reviewer #2. In you revision please address the comments/suggestions made by reviewer #2 points 1-4. I may suggest that since you have provided in the Supporting Information the mathematical proofs for the discrete and continuous model, it would be interesting if in the 'Results and Discussion' part of the manuscript you will include more ecological explanations based on your model results in relation to link between phenotypic variability in ecological traits and the functional aspect of biodiversity in marine microbes; and less detailed mathematical explanations if possible.    

We look forward to receiving your revised manuscript.

Kind regards,

Andrea Belgrano, Ph.D.

Academic Editor

PLOS ONE

Journal Requirements:

Reviewers' comments:

Reviewer's Responses to Questions

**Comments to the Author**

1. Is the manuscript technically sound, and do the data support the conclusions?

Reviewer #1: Partly

Reviewer #2: Partly

2. Has the statistical analysis been performed appropriately and rigorously? 

Reviewer #1: N/A

Reviewer #2: Yes

3. Have the authors made all data underlying the findings in their manuscript fully available?

Reviewer #1: Yes

Reviewer #2: Yes

4. Is the manuscript presented in an intelligible fashion and written in standard English?

Reviewer #1: No

Reviewer #2: Yes

5. Review Comments to the Author

Reviewer #1: No review provided.

Reviewer #2: This paper presents a game-theory based mathematical proof that multiple species can coexist in single niche, as long as each has individuals that vary in their competitive abilities. This is a very interesting and current topic, as the field of microbial ecology tries to explain high diversity and functional redundancy in natural communities.

As I read the paper, I had a number of questions about the ecological/evolutionary robustness of how the model is structured. These questions may simply reflect my ignorance in the area of game theory (a topic I’m not capable of evaluating) but they may indicate areas that need to be further explained or justified for the non-modelers among us.

1. Figure 2 had me stumped. I may be taking it too literally, but the makeup of the individuals that compete in round 1 doesn’t reflect the winners of round 2. In the model, how do cells with new colors arise (e.g., a dark brown cell newly appears in Round 2), and why didn’t they have to compete in Round 1? I understand that there aren’t actually any modeled cells in the mathematical solutions, but from a conceptual perspective, can this happen?

2. Figure 2, again: In the model, are all cells competed with all other cells in a given round (i.e., are all pairwise competitions occurring), or is each cell randomly assigned to just one pairwise competition? The latter seems ecologically unrealistic in a community milieux. How does the mathematical proof handle this?

3. My understanding is that the cells get new random assignments of fitness, justified by potential changes in fitness due to noise, plasticity, or genetic differences. But genetic differences aren’t erased in the next generation; even short-term DNA methylation has a lifetime of several generations. And what determines plasticity if the capability for it isn’t encoded in the genome? I am on board with noise, but it seems like there would need to be an incredible amount of physiological noise for this to drive the model’s fitness heterogeneity at the necessary level.

4. If you haven’t seen this paper, you may want to take a look. Despite the title, it has nice experimental evidence of phenotypic heterogeneity in bacterial species traced to differences in mRNA transcription. https://science.sciencemag.org/content/371/6531/eaba5257.abstract?casa_token=9mGqrOddhrUAAAAA:iRzo9Zy5yBMFScOp5CvmiDxv5ysg63TZ1eljmCBQOiIzAOCVlLuYbrKTjqfaN4GALkLCHoh5jawmkbiJ

6. PLOS authors have the option to publish the peer review history of their article (what does this mean?). If published, this will include your full peer review and any attached files.

Reviewer #1: No

Reviewer #2: No

---

## [Author Response · Author response to Decision Letter 0]

27 Jun 2021

Dear Editors, 

We thank the editor and reviewers for their time and feedback. We agree with the comments from both reviewers and the editor. We value the input and believe it has strengthened our contribution.

1. (Author reply to editor's comments): Thank you for handling and reviewing our paper. We have addressed the points by reviewer 2 below. In addition, we have revised the results and discussion section by moving the auxiliary mathematical results to the supplemental material and splitting into two sections. The results contain the main theorem that has been re-written to include the corollary, and we have included additional ecological explanations of these results. These results are followed by a discussion section dedicated to the extended ecological and evolutionary implications. Here we focused the discussion on 1) comparison of different competitive strategies among microbes, 2) the role of phenotypic heterogeneity in community composition (or competition) and 3) future directions to better conceptualize competition in microbial systems. We hope this addresses your concerns.

2. (Author reply to Reviewer 2 general comments): Thank you for your effort in reviewing our paper, your constructive feed-back, and your pertinent questions. The model was only briefly described, as it utilizes a prior model (Menden-Deuer & Rowlett 2014). That said, any paper should be self-explanatory and we have revised all sections and figures, so that hopefully your questions are addressed (see below). 

2.1. (Reviewer comment) Figure 2 had me stumped. I may be taking it too literally, but the makeup of the individuals that compete in round 1 doesn’t reflect the winners of round 2. In the model, how do cells with new colors arise (e.g., a dark brown cell newly appears in Round 2), and why didn’t they have to compete in Round 1? I understand that there aren’t actually any modeled cells in the mathematical solutions, but from a conceptual perspective, can this happen?

2.1 (Author reply) That is a great question and important for us to clarify. The colors are arbitrary and are used to show lots of diversity in the population. The round 1 competitors are a small and arbitrary subset of all possible, highly variable cells in that population. So, the dark brown cell in Round 2 was already part of the population in Round 1. Success in one round of competition though does not translate to the next round. This reflects the fact that our model considers all traits simultaneously, and these can and do change. If an individual has an advantageous behavior at one round of competition (e.g. to acquire nutrients), it has a high competitive ability. If in the next round of competition, this behavior is no longer advantageous (e.g. a predator comes along and the individual should flee), then the competitive ability in this next round is lower. Even if it is the same individual doing the same thing. Or, for example, an individual can change its behavior, without the environment changing, so that its behavior is more or less advantageous. It doesn’t change the individual’s genetic makeup. We address this further in response to point 3. Here we have revised the figure legend and explanation associated with the figure.

Figure 2 legend revision now reads: 

Phenotypically variable individuals compete. At each round of competition, a subset of diverse individuals from a large cohort competes and the outcomes of competition are assessed based on their relative competitive abilities (values within cells). Supported by empirical observations, the competitive ability of clonal individuals can be expressed heterogeneously in identical conditions and vary over time. Our work reflects this by assigning individuals their competitive ability according to the strategy of the species. Thus, variability amongst individuals and the strategy of the species is preserved.

2.2. (Reviewer comment) Figure 2, again: In the model, are all cells competed with all other cells in a given round (i.e., are all pairwise competitions occurring), or is each cell randomly assigned to just one pairwise competition? The latter seems ecologically unrealistic in a community milieux. How does the mathematical proof handle this?

2.2 (Author reply) Thanks for letting us clarify this. Each round of competition entails a pair wise competition between two cells, and then cells sequentially compete with others. Our model and theory rely on discrete interactions between two individual cells. This is a key element that rather than considering competition as a ‘population average’ we discretize the interaction to how they occur individually. This is built into the mathematical equations. Hopefully our revisions of the explanation around Figure 2 mentioned above also helped with this issue.

In addition, and we have amended the description of the model simulations based on your comments in lines 165-167: 

We view this CA as a cumulative trait because a single competitive interaction does not typically yield as decisive an outcome as division or death (although it can). Thus, our competitions reflect the cumulative outcomes of competitions over an organism’s generation.

2.3. (Reviewer comment) My understanding is that the cells get new random assignments of fitness, justified by potential changes in

fitness due to noise, plasticity, or genetic differences. But genetic differences aren’t erased in the next

generation; even short-term DNA methylation has a lifetime of several generations. And what determines

plasticity if the capability for it isn’t encoded in the genome? I am on board with noise, but it seems like there

would need to be an incredible amount of physiological noise for this to drive the model’s fitness heterogeneity at the necessary level.

2.3 (Author reply) Yes, there would be an incredible amount of noise and the presence of that noise is empirically verified, as the many citations in the introduction show, including the Kuchina et al. 2021 paper to which you kindly brought our attention. Virtually every time sampling occurs at the individual level, phenotypic variability is present at high magnitude, at the individual cell level (e.g. every cell sampled is different). And yes, the cells get new assignments of fitness, but they are not completely random. The assigned fitness is chosen at random from within the distribution defined for the species (e.g. invariant, uniform…). For example, the invariant distribution has no variation or noise amongst individuals at all, they are all perfect clones. This approach of choosing the assigned fitness from the distribution defined for the species is supported by the fact that clonal individuals can yet exhibit phenotypical variability, irrespective of environmental conditions, meaning even in identical environmental conditions, clonal individuals can have different metabolic and behavioral functions [Bruijning et al.(2020)]. Of course, environmental heterogeneity, a cells physiological history and any other short term stimulus would only serve to further increase this phenotypic heterogeneity. We clarify this in the revised manuscript by expanding the section “Simulations for the discrete model” to give the specific justifications for our formulations. This is contained in lines 180-191 of the revised manuscript, and for your convenience, we copy the explanation here: 

The seemingly infinite variation in traits empirically observed (see citations in the introduction) is implemented in this model by assigning cohorts of individuals (populations or species) infinite variability as characterized by the uniform distribution. The ramification of this population-level competitive strategy is contrasted with other commonly used strategies, such as a biomodal distribution or an invariant distribution (lacking variance represented by a single, constant competitive ability). Heterogeneous expression of traits even by clonal individuals (Bruijning et al. 2020) in identical environmental conditions is manifested in the model by allowing random assignments of competitive abilities (CA) to individuals, and allowing those to change over time, irrespective of prior successes of that individual. Ecologically, this reflects conditions where one trait may be advantageous in one condition but disadvantageous in another condition. Each competition is evaluated discretely between two individuals.

2.4. (Reviewer comment) If you haven’t seen this paper, you may want to take a look. Despite the title, it has nice experimental

evidence of phenotypic heterogeneity in bacterial species traced to differences in mRNA transcription.

https://science.sciencemag.org/content/371/6531

/eaba5257.abstract?casa_token=9mGqrOddhrUAAAAA:iRzo9Zy5yBMFScOp5CvmiDxv5ysg63TZ1eljmCBQOiIzAOCVlLuYbrKTjqfaN4GALk

2.4 (Author reply) You are right, this is a great paper and certainly one we would have missed based on the title. We incorporated a citation of this work in the manuscript, as further evidence of persistent and ubiquitous phenotypic heterogeneity in asexually reproducing microbes. We have added the reference as well as the following statement to the text in lines 13-15: 

Recent methodological break throughs have allowed the discovery of heterogeneity at the level of gene expression activation and revealed rare sub-populations in bacteria (Kuchina et al. 2021).

We thank you for your time and consideration, 

Julie Rowlett, on behalf S. Menden-Deuer, M. Nursultanov, S. Collins, and T. A. Rynearson

Corresponding authors contact information:

Susanne Menden-Deuer Julie Rowlett

Professor of Oceanography Associate Professor, Dept of Mathematics

email: smenden@uri.edu Email: julie.rowlett@chalmers.se

Phone: +401 218 5466 Phone: +46 732006949

Address: University of Rhode Island Mathematical Sciences

Graduate School of Oceanography Chalmers University

Narragansett, RI 02882, USA 41296 Gothenburg, Sweden

---

## [Editor Report · Decision Letter 1]

5 Jul 2021

Biodiversity of marine microbes is safeguarded by phenotypic heterogeneity in ecological traits

PONE-D-21-01716R1

Dear Dr. Rowlett,

We’re pleased to inform you that your manuscript has been judged scientifically suitable for publication and will be formally accepted for publication once it meets all outstanding technical requirements.

Kind regards,

Andrea Belgrano, Ph.D.

Academic Editor

PLOS ONE

Additional Editor Comments (optional):

Thank you for addressing all the comments made by reviewer #2, and providing clarity.

---

## [Editor Report · Acceptance letter]

13 Jul 2021

PONE-D-21-01716R1 

Biodiversity of marine microbes is safeguarded by phenotypic heterogeneity in ecological traits 

Dear Dr. Rowlett:

I'm pleased to inform you that your manuscript has been deemed suitable for publication in PLOS ONE. Congratulations! Your manuscript is now with our production department. 

Kind regards, 

on behalf of

Dr. Andrea Belgrano 

Academic Editor

PLOS ONE